# Study on Adaptability of Test Methods for Workability of Fresh Self-Compacting SFRC

**DOI:** 10.3390/ma14185312

**Published:** 2021-09-15

**Authors:** Xinxin Ding, Haibin Geng, Kang Shi, Li Song, Shangyu Li, Guirong Liu

**Affiliations:** 1International Joint Research Lab for Eco-Building Materials and Engineering of Henan, School of Civil Engineering and Communication, North China University of Water Resources and Electric Power, Zhengzhou 450045, China; dingxinxin@ncwu.edu.cn (X.D.); hbgeng@stu.ncwu.edu.cn (H.G.); Z201910311281@stu.ncwu.edu.cn (K.S.); lishangyu@ncwu.edu.cn (S.L.); 2Collaborative Innovation Center for Efficient Utilization of Water Resources, North China University of Water Resources and Electric Power, Zhengzhou 450045, China; liugr@ncwu.edu.cn; 3Yellow River Institute of Hydraulic Research, Zhengzhou 450003, China

**Keywords:** self-compacting SFRC, workability, test methods, modification, filling ability, passing ability, stability

## Abstract

To ensure the quality of concrete construction, the workability of fresh mix measured by rational test methods is critical to be controlled. With the presence of steel fibers, whether the test methods and evaluation indices of fresh self-compacting concrete (SCC) are adaptable for self-compacting steel fiber reinforced concrete (SFRC) needs to be systematically verified. In this paper, seven groups of self-compacting SFRC, referenced with one group SCC, were prepared by using the mix proportion design method based on the steel fiber-aggregates skeleton packing test. The main factors included the volume fraction and the length of hooked-end steel fiber. Tests for filling ability, passing ability, and stability of fresh self-compacting SFRC and SCC were carried out. Results indicate that the adaptability was well for the slump-flow test with indices of slump flow and flow time *T*_500_ to evaluate the filling ability, the J-ring flow test with an index of *PA* level to evaluate the passing ability, and the static segregation test with an index of static segregation resistance to evaluate the stability of fresh self-compacting SFRC. By the repeated tests and measurements, the slump cone should be vertically lifted off to a height of 300 mm within 3 s at a constant speed, the spacing of the rebar in the J-ring test should be adjusted to be two times the fiber length. If the table jumping test is used, the dynamic segregation percent should be increased to 35% to fit the result of the static segregation test. Good workability of the self-compacting SFRC prepared in this study is presented with the general evaluation of test results.

## 1. Introduction

With the development of concrete construction technology, self-compacting concrete (SCC) has been widely applied in engineering [1,2]. Combined with the high-performance peculiarity of steel fiber reinforced concrete (SFRC), the self-compacting SFRC was also developed [3,4,5]. Normally, the workability of SCC is evaluated by the filling ability, passing ability, and stability [1,2,6,7,8]. Meanwhile, SCC is detected with high plastic viscosity and low yield stress to ensure good placeability and high segregation resistance [9]. Compared with SCC, the workability of self-compacting SFRC changes in different aspects due to the mix rheology was affected significantly by the addition of steel fibers, and the fiber segregation decreases with the increased plastic viscosity [9,10,11,12]. This is not only affected by the morphology of steel fibers, including the surface roughness and geometry, but also by the fraction of steel fiber by volume of SFRC [12,13,14,15,16]. In view of the effect of steel fiber, the applicability of the test methods for SCC needs to be verified for self-compacting SFRC.

Learning from the studies on the rheology and workability of mortar with the straight steel fiber length of 13 mm and an aspect ratio of 65, the proper content of straight fibers would not reduce the flowability and plastic viscosity of mixture and even increase the flow spread and flow rate due to the broken flocculation and net structure in mortar by steel fibers [12,13]. When the fiber factor became close to a certain critical value, the contact network interlocked and tangled, and even the balling of fibers began to govern the rheology. Compared with the flow rate, the flow spread is more applicable to illustrate the influence of fibers on the rheology of fresh mortar. Studies on the rheology and workability of fresh self-compacting SFRC indicated that both the yield stress and plastic viscosity of self-compacting SFRC increase and workability decreases with the increase in fiber volume fraction; no matter what type of the steel fibers, straight, crimped, and hooked-end, only the length of hooked-end steel fiber has the significant influence on the plastic viscosity. The effect of the deterioration of the slump-flow and flow time *T*_500_ is small with the addition of fiber less than 1% volume fraction [14,15]. In addition, the plastic viscosity was decreased by the appropriate amount of straight steel fiber with a length of 6 mm and an aspect ratio of 37.5 [12,13,14,15]. Meanwhile, the plastic viscosity and the yield stress of fresh self-compacting SFRC linearly increased with the volume fraction of hooked-end steel fiber with a length of 35 mm and an aspect ratio of 65. When half of the volume fraction of steel fiber changed to the hooked-end steel fiber with a longer length of 50 mm and an aspect ratio of 80, the yield stress increased, accompanied by the reduced plastic viscosity. The flow time tested by V-funnel showed a linear relationship with the plastic viscosity, while there was no clear tendency between the slump flow with the yield stress or the plastic viscosity in self-compacting SFRC [16].

Although test methods for the workability of self-compacting SFRC are specified in codes [3,17], few studies were focused on the adaptability of these methods and the reasonable evaluation indexes corresponding to the test results. A study on the workability of self-compacting SFRC showed that the slump flow test could be used to evaluate the flowability, and the J-ring and L-box tests present similar results of the passing ability of self-compacting SFRC [18]. In this condition, the current specifications lack a solid research basis for the determination of workability of self-compacting SFRC by using the test methods for SCC. Therefore, a study on this issue is of importance for the construction quality of self-compacting SFRC.

In this paper, the study focused on the test methods of workability, including the filling ability, the passing ability, and the stability of fresh self-compacting SFRC. Seven groups of self-compacting SFRC referenced with one group SCC were prepared, considering the main factors with the volume fraction and the length of hooked-end steel fiber. The lift-off height of the slump cone and the flow time *T*_500_ and final time *T*_final_ in the slump-flow test were discussed in the filling ability. The rebar spacing of the J-ring test and evaluation indexes considering fiber length were considered for the passing ability. The static segregation test and the dynamic segregation test with different detection buckets were carried out to comparatively analyze the suitable indexes for the stability of self-compacting SFRC. Based on the test results and considering the features of steel fiber, the applicability of current test methods is evaluated, and the adjustment of details are proposed.

## 2. Experimental Work

### 2.1. Raw Materials

Ordinary silicate cement in the strength grade of 42.5, fly ash of class-II, and slag powder were used as binders. The physical and mechanical properties of cement are presented in Table 1. The density and water demand of fly ash were 2350 kg/m^3^ and 83%. Their properties met the relevant specifications of China codes GB 175, GB/T 1596, and GB/T 51,003 [19,20,21].

The fine aggregate was the manufactured sand with a fineness modulus of 2.73 and a stone powder content of 7.3%. The coarse aggregate was crushed limestone with a maximum particle size of 16 mm and continuous particle grading. The basic physical and mechanical properties of the aggregates are presented in Table 2, which met the specifications of China codes GB/T 14684 and GB/T 14685 [22,23].

Hooked-end steel fiber was used with the length of 25.1, 29.8, and 34.8 mm, and the diameter of 0.5 mm. The aspect ratio was 50, 60, and 70, respectively. The tensile strength was 1150 MPa. The measurements were conducted according to the specification of China code JG/T472 [17]. They were marked as HFa, HFb, and HFc successively.

The mixing water was tap water. The polycarboxylate superplasticizer with a water-reducing ratio of up to 30% and solid content of 30% and were produced by Jiangsu Sobute New Materials Co. Ltd. of China (Nanjing, China).

### 2.2. Mix Proportion

The mix proportion was designed by using the design method of absolute volume based on the packing test of steel fiber-aggregates skeleton [24,25,26]. For the self-compacting SFRC with HFb steel fiber, the volume fraction was from 0.4% to 1.6%. For the self-compacting SFRC with HFa and HFc steel fiber, the volume fraction was 1.2%. The mix proportion is presented in Table 3.

### 2.3. Test Methods

The workability of fresh mixes was measured in accordance with the specification of China code JGJ/T283 [1], which is identical to ASTM C1611 [6] and ASTM C 1621 [7].

The filling ability was evaluated by the slump-flow test. According to the specification of China code JGJ/T283 [1], the slump cone should be lifted off to a height of 300 mm within 2 s. However, according to the specification of China code CECS13 [3], the slump cone should be lifted off within 5 s while no requirement of the lift-off height is defined. Therefore, considering the height of the discharge outlet of SCC above the casting surface being less than 1.0 m, the lift-off height of the slump cone was set as 300 and 600 mm. After the repeated operation to lift the slump cone at a constant speed, the lifting time was determined to be within 3 s. Normally, the indices are the diameter of slump-flow and the time of slump-flow to a diameter of 500 mm (*T*_500_). Based on some results reported [6,27,28], the final time (*T*_final_) of slump-flow at the stopping state can reflect the filling performance of the mixture. This was also detected in this study.

The passing ability was evaluated by the J-ring test compared to the slump-flow test with the index of the difference of diameter between slump-flow and J-ring flow (*PA*). According to China code JGJ/T283 [1], the J-ring is 300 mm in diameter with rebar of 16 mm and spacing of 58.9 mm. The net spacing of 42.9 mm meets the requirement that is no less than two times of maximum particle size of coarse aggregate for SCC, due to the maximum particle size of coarse aggregate for SCC is limited to 20 mm. According to China code CECS13 [3], the J-ring is 300 mm in diameter with rebar of 10 mm and spacing of 48 ± 2 mm. The spacing could be adjusted due to the application required to be one to three times the fiber length. To get the equivalent passing ability of self-compacting SFRC to SCC, the rebar spacing was made as two times the fiber length. The modified J-ring is presented in Figure 1.

The stability of SCC was evaluated by the static segregation test with an index of the static segregation percent (*SR*) and the dynamic segregation test with an index of the dynamic segregation percent of coarse aggregate (*f*_m_). However, a different detection bucket is used in China codes. According to China code JGJ/T283 [1], the detection bucket is 115 mm in inner diameter and 300 mm inheight composed of three sections; each section is 100 mm in height. According to China code CECS13 [3], the detection bucket is 150 mm in inner diameter and 450 mm in height composed of three sections; each section is 150 mm in height. In this study, the two detection buckets were used for the dynamic segregation test on a jump table.

## 3. Test Results and Discussion

### 3.1. Filling Ability

The test results of the slump-flow with lift-off height of 300 and 600 mm changed with the fiber factor and are presented in Figure 2. Due to different lengths of hooked-end steel fiber tested in this study, the fiber factor (*λ*_f_) is the product of the aspect ratio multiplied by the volume fraction of steel fiber and is applied to reflect the comprehensive effect of steel fiber on workability in the following discussion.

Compared to those with a lift-off height of 300 mm, the slump-flow with a lift-off height of 600 mm of fresh SCC enlarged 35 mm; and those of fresh self-compacting SFRC enlarged about 20–75 mm. This indicates the slump-flow was influenced by the kinetic potential during the lift-off. In the same 3 min period of lift-off time, the kinetic potential increased at a lift-off height of 600 mm compared to that at a lift-off height of 300 mm, and the slump-flow was increased. Meanwhile, a similar reduction in slump-flow existed with the increase in fiber factor at the two lift-off heights. This is due to the decrease in flowability of the fresh matrix with the enhanced bridge effect of steel fiber in the binder paste.

As presented in Figure 3, compared to those with a lift-off height of 300 mm, the flow time *T*_500_ and *T*_final_ with a lift-off height of 600 mm of fresh SFRC was prolonged by about 1 and 6 s, respectively. The HFb16 presented a greater decrease in slump-flow with not much elongation in flow time. With the fiber length increased to be 34.8 mm, the slump-flow decreased, with the longest flow time of 6.7 s. No obvious relationship existed between flow time and fiber factor. This indicates that the rational mix proportion modified the flowability of fresh mixtures with the less blocking effect of steel fibers [25,26]. Generally, the dispersion of *T*_500_ was smaller than that of *T*_final_. Therefore, *T*_500_ is a rational index evaluating the filling ability of self-compacting SFRC.

Generally, a lift-off height of 300 mm in 3 s for the slump cone is adaptable to be used for the controlling parameter of the test operation. Since the standard height of the slump cone is 300 mm, the lifting height is equal to the height of the slump cone, which is convenient for intuitive comparison control. In this condition, as presented in Figure 4, the slump-flow of self-compacting SFRC is 560–685 mm with a flow time *T*_500_ of 4.33–6.71 s. The degree of slump-flow met *SF*1 or *SF*2, and the level of flow time met *VS*1.

### 3.2. Passing Ability

The J-ring flow and passing ability level of fresh mixtures tested with standard rebar spacing are presented in Figure 5. The J-ring flow of fresh SCC was 650 mm with a *PA* value of 5 mm, and the passing ability was *PA*2 level. With the presence of steel fiber, the J-ring flow decreased. For the self-compacting SFRC with a volume fraction of 1.6% (HFb16), the J-ring flow decreased sharply to 500 mm with a larger PA value of 95 mm. Meanwhile, a much larger decrease in the J-ring flow and increase in *PA* value happened on the self-compacting SFRC with a fiber length of 34.8 mm (HFc12). This indicates the blocking effect of steel fiber on passing ability increased with the increase in volume fraction and fiber length.

With the modified rebar spacing of the J-ring, the J-ring flow increased for all self-compacting SFRC, the *PA* values were all below 20 mm, as presented in Figure 6. At the same time, the height difference of mixture inside to outside of the J-ring is presented in Table 4. Compared to those of fixed rebar spacing of J-ring, the height difference of modified rebar spacing had a less increase with the increase in volume fraction and fiber length. Therefore, the rebar spacing modified to two times the fiber length is much more rational for evaluating the passing ability of fresh self-compacting SFRC. Furthermore, considering that longer fiber is not always used for self-compacting concrete, due to the more deterioration of passing ability and filling ability and even mechanical properties [15], rebar spacing of 70 mm is advised for self-compacting SFRC.

### 3.3. Segregation Resistance

The test results of the static segregation percent of fresh mixtures are presented in Figure 7. The static segregation resistance of fresh SCC was over the requirement, while all of fresh self-compacting SFRC met the *SR*1 level. With the volume fractions of 1.4% and 1.6%, and a fiber length of 25.1 mm, the static resistance of fresh self-compacting SFRC met *SR*2 level. This indicates that the plastic viscosity of the fresh mixture increased with the increase in volume fraction, which is consistent with the [9], and the fiber length of 25.1 mm matched the maximum particle size of 20 mm.

The test results of dynamic segregation percent by two sizes of detection buckets are presented in Figure 8. Generally, the dynamic segregation was larger by using the detection bucket of CECS13:2009 than by using the detection bucket of JGJ/T283. This is due to the decreased confinement of a detection bucket with a larger diameter on coarse aggregate and steel fiber. Meanwhile, the dynamic segregation of steel fiber was basically higher than that of the coarse aggregate. This is due to differing from the coarse aggregate surrounded by binder paste; the elongated steel fiber with high self-weight has the potential trend of layering from the matrix under dynamic action. However, the segregation of steel fiber differed a little from that of the coarse aggregate tested with the same detection bucket, the dynamic segregation of fresh self-compacting SFRC can be evaluated by the coarse aggregate to simplify the detection. In view of the convenience of test operation, the smaller detection bucket used in China code JGJ/T283 is rational for the test of dynamic segregation of self-compacting SFRC.

The test results of dynamic segregation percent of coarse aggregate by using the method of China code JGJ/T 283 [1] are presented in Figure 9. Only HF16 met the requirement of less than 10%; this requirement may be hypercritical. With the prevailing principle of static segregation resistance, the dynamic segregation percent of coarse aggregate in fresh self-compacting SFRC could be taken as 35%.

## 4. Conclusions

In this paper, which was aimed at building applicable test methods and rational evaluation indexes of fresh self-compacting SFRC, seven groups of self-compacting SFRC with different volume fractions and lengths of hooked-end steel fiber were taken as test objects. The test methods of the filling ability, the passing ability, and the stability were carried out. Conclusions can be drawn as follows:(1)The filling ability of self-compacting SFRC can be determined by the slump-flow test and evaluated with the indices of slump-flow and flow time *T*_500_. The slump cone should be vertically lifted off to a height of 300 mm within 3 s at a constant speed.(2)The passing ability of self-compacting SFRC can be determined by the J-ring test and evaluated with an index of *PA* level. The rebar spacing of the J-ring should be adjusted to be two times the fiber length to get an equivalent evaluation to the passing ability of SCC.(3)The stability of self-compacting SFRC can be determined by the static segregation test of the coarse aggregate. When the dynamic segregation test is used, the dynamic segregation percent should be adjusted to be 35%. If the limit of 10% as specified in the China codes is adopted, it is hypercritical to self-compacting SFRC, even SCC, the evaluation of dynamic segregation resistance will be a contradiction to the evaluation of static segregation resistance.(4)The self-compacting SFRC prepared in this study has good workability with filling, passing, and stable abilities. The mix proportion of self-compacting SFRC was rationally designed by using the direct mix design method based on the packing test of the steel fiber-aggregates skeleton. However, much more test data should be accumulated to verify the suggested adjustments of details for the rational specifications.

## Figures and Tables

**Figure 1 materials-14-05312-f001:**
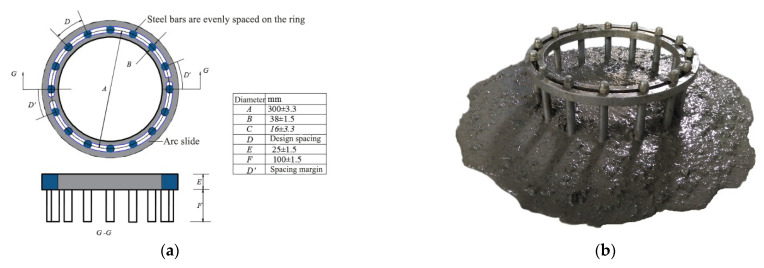
Modified J-ring with variable rebar spacing: (**a**) diagrammatic sketch; (**b**) picture.

**Figure 2 materials-14-05312-f002:**
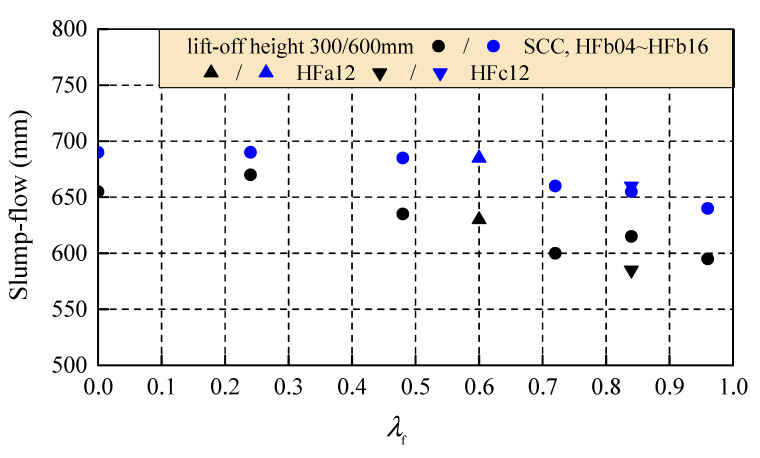
Slump-flow with different lift-off heights changed with the fiber factor.

**Figure 3 materials-14-05312-f003:**
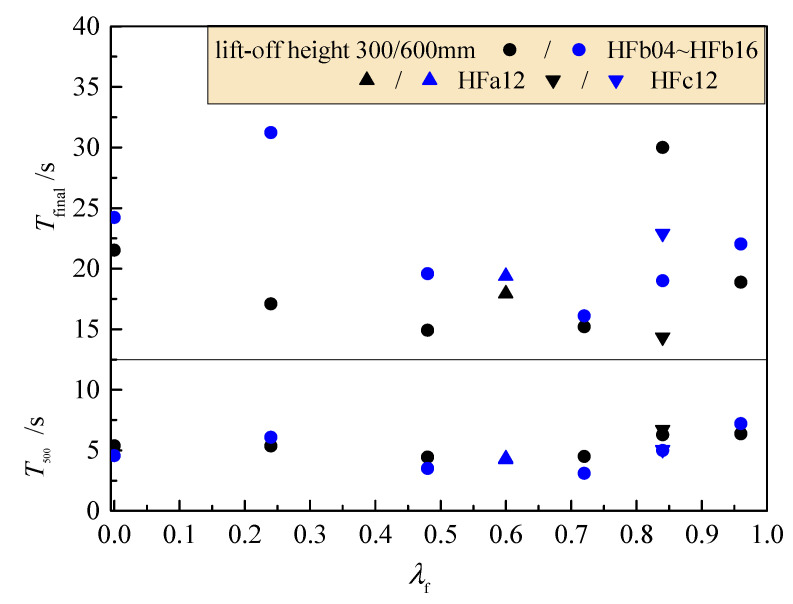
Flow time *T*_500_ and *T*_final_ with different lift-off heights changed with the fiber factor.

**Figure 4 materials-14-05312-f004:**
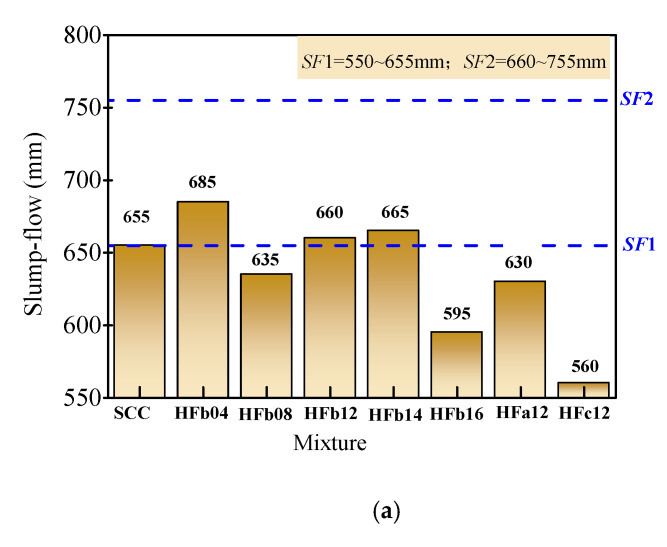
Filling ability evaluated by slump-flow and flow time. (**a**) Slump-flow. (**b**) Flow time *T*_500_.

**Figure 5 materials-14-05312-f005:**
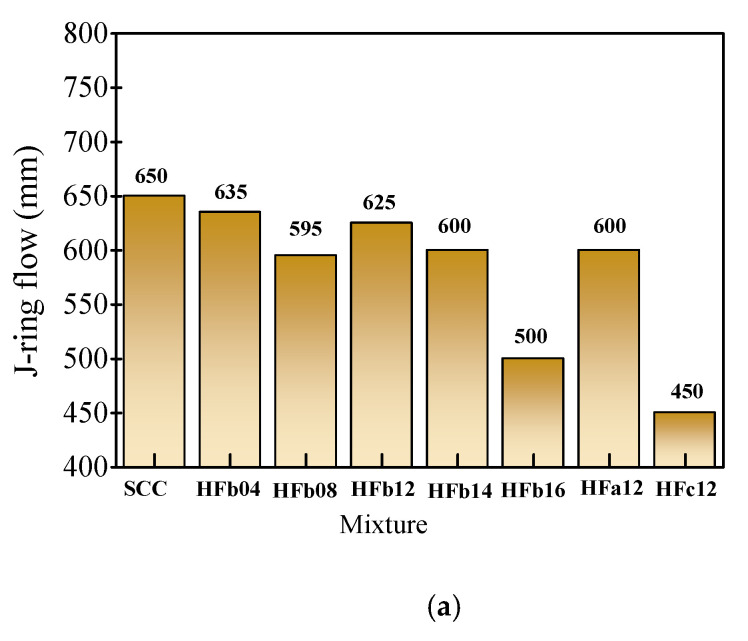
Test results and the corresponding performance level of passing ability. (**a**) J-ring flow. (**b**) *PA*.

**Figure 6 materials-14-05312-f006:**
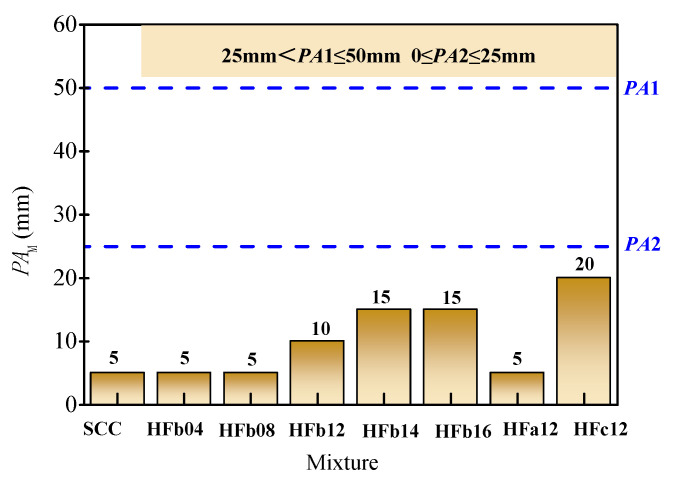
Passing ability of fresh mixtures tested by using modified rebar spacing of the J-ring.

**Figure 7 materials-14-05312-f007:**
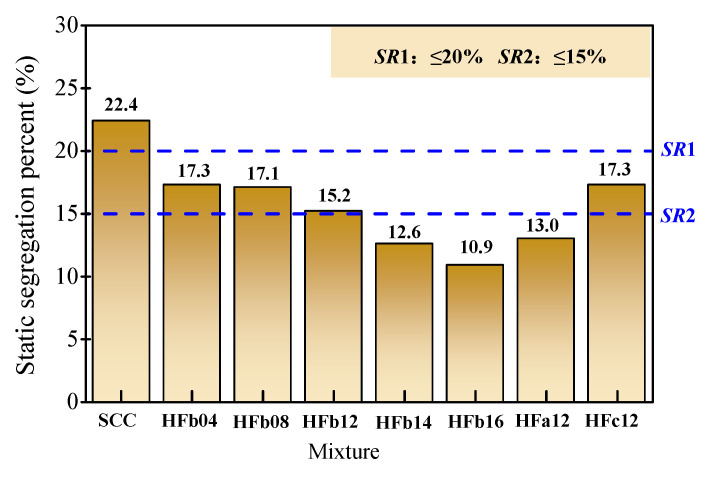
Test results and the corresponding performance level of static segregation resistance.

**Figure 8 materials-14-05312-f008:**
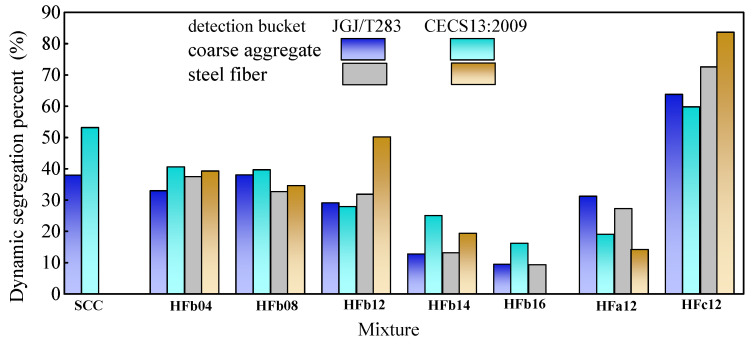
Test results of dynamic segregation percent of mixtures by using different buckets.

**Figure 9 materials-14-05312-f009:**
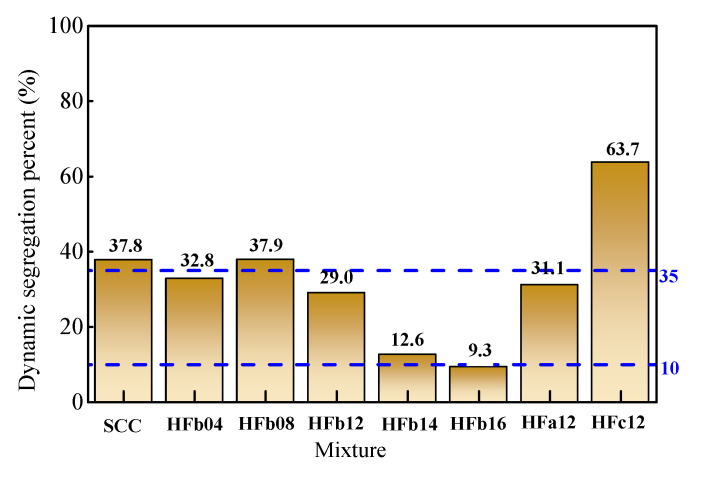
Dynamic segregation percent and resistance level of coarse aggregate.

**Table 1 materials-14-05312-t001:** Basic physical and mechanical properties of cement.

Fineness (%, 45 μm)	Water Requirement of Normal Consistency (%)	Density (kg/m^3^)	Setting Time (min)		Compressive Strength (MPa)		Flexural Strength (MPa)	
			Initial	Final	3 d	28 d	3 d	28 d
8.6	26.6	3195	176	222	34.9	53.8	6.23	8.83

**Table 2 materials-14-05312-t002:** Basic physical and mechanical properties of aggregates.

Name	Apparent Density (kg/m^3^)	Bulk Density (kg/m^3^)	Closing Packed Density (kg/m^3^)	Crush Index (%)	Mud Content (%)	Flat and Needle Particles Content (%)
Manufactured sand	2740	1620	1840	-	-	-
Crushed stone	2735	1530	1590	12.2	0.5	5.25

**Table 3 materials-14-05312-t003:** Mix proportion of self-compacting SFRC by using the direct mix design method.

Mixture		SCC	HFb04	HFb08	HFb12	HFb14	HFb16	HFa12	HFc12
Water to binder ratio *w/b*		0.31	0.31	0.31	0.31	0.31	0.31	0.31	0.31
Sand ratio *β*_s_ (%)		50	52	54	56	57	58	55	57
Fly ash contnet (%, by mass)		30	30	30	30	30	30	30	30
Steel fiber	type	-	HFb	HFb	HFb	HFb	HFb	HFa	HFc
	*v*_f_ (%)	0	0.4	0.8	1.2	1.4	1.6	1.2	1.2
Water (kg/m^3^)		192	201	210	219	223	228	214	223
Cement (kg/m^3^)		433	454	474	494	504	514	484	504
Fly ash (kg/m^3^)		186	194	203	214	216	220	207	216
Crushed stone (kg/m^3^)		751	675	601	527	491	455	553	502
Manufactured sand (kg/m^3^)		751	763	774	783	788	792	784	782
Steel fiber (kg/m^3^)		0	31.4	62.8	94.2	109.9	125.6	94.2	94.2
Water-reducer (kg/m^3^)		5.57	5.51	5.42	5.30	5.40	5.51	5.19	5.40

**Table 4 materials-14-05312-t004:** The height difference of fresh concrete between inside and outside of the J-ring.

Mixture	SCC	HFb04	HFb08	HFb12	HFb14	HFb16	HFa12	HFc12
*HD* (mm)	0	23	30	26	42	62	25	46
*HD*_M_ (mm)	0	0	0	6	25	26	21	23

## Data Availability

The data presented in this study are available on request from the corresponding author.

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
