# Peer review of "Study on Adaptability of Test Methods for Workability of Fresh Self-Compacting SFRC"

_materials, 2021, doi:10.3390/ma14185312_

Round 1
Reviewer 1 Report
The document discusses results concerning the workability of steel fiber reinforced self compacting concrete and also the rearrangement of protocols fot the evaluation of several parameters. The document is well-written, the aim well-delineated and the conclusions clear and well-supported by data.
Only few minor issues need to be considered before the publication.
1- Line 47, the aspect ratio is dimensionless
2- Line 115, "identical to" instead of "identical with"
3- Lines 141-142, "dynemic" typo
4- Lines 156-158, convoluted sentence
5- Figure 2, use the same legend style as Figure 3
6- Concerning the dispersion of data (Line 173), is it consequence of the mix properties or it is just experimental noise? Do the authors verify repeatability of tests?
7- Figures 4, 5, 6, 7, and 9, using curly brakets on the right hand side of the plots allows to delete text in plot boxes, leading the figures to be cleaner and more elegant.
Anyway, the legend styles change between figures. Authors are recommended to use an editing style as uniform as possible.
Author Response
Manuscript number: Materials-1378920
MS Type: Article
Title: Study on adaptability of test methods for workability of fresh self-compacting SFRC
Correspondence Author: Xinxin Ding
Dear editor,
Thanks very much for your attention and the reviewers’ evaluation and comments on paper Materials-1378920. Those comments are all valuable and very helpful for revising and improving our paper, as well as the important guiding significance to our researches. Here, we have made extensive modification on the original manuscript, and carefully proof-read the manuscript to minimize typographical, grammatical, and bibliographical errors. The main revised parts are represented as Red. We also attached revised manuscript in the format of MS word for your approval. we have presented the original and modified figures 4, 5, 6, 7, and 9 in parallel in the revised manuscript. In our opinion, the original figures are more concise and explicit, and the modified figures are cleaner and more elegant. All of them are OK for us. So, please help us make a judgment. Here below is our description on revision according to the reviewer 1’ comments.
Comment 1:Line 47, the aspect ratio is dimensionless
Response:It has been revised.
Comment 2: Line 115, "identical to" instead of "identical with"
Response: It has been revised.
Comment 3: Lines 141-142, "dynemic" typo
Response: It has been revised.
Comment 4- Lines 156-158, convoluted sentence
Response: It has been revised.
Comment 5- Figure 2, use the same legend style as Figure 3
Response: It has been revised.
Comment 6- Concerning the dispersion of data (Line 173), is it consequence of the mix properties or it is just experimental noise? Do the authors verify repeatability of tests?
Response: It has been revised.
Comment 7- Figures 4, 5, 6, 7, and 9, using curly brakets on the right hand side of the plots allows to delete text in plot boxes, leading the figures to be cleaner and more elegant.
Response: Thanks for your kind reminder. we have presented the original and modified figures in parallel in the revised manuscript. In our opinion, the original figures are more concise and explicit, and the modified figures are cleaner and more elegant. All of them are OK for us. So, please help us make a judgment.
Comment 8-Anyway, the legend styles change between figures. Authors are recommended to use an editing style as uniform as possible.
Response: Thanks for your kind remind. We have revised the legend styles as uniform as possible.
Reviewer 2 Report
The proposed manuscript exposes a Study on adaptability of test methods for workability of fresh self-compacting SFRC (steel fiber reinforced concrete). Seven concrete were prepared and the fresh properties were tested (slum flow, J-ring) and some recommendations are then deduced ‘slump cone should be vertically lifted off to height of 300 mm within 3 s at constant speed, the spacing of rebar in J-ring test should be adjusted to be 2 times of fiber length.’
The article is rather short and composed of four parts: an introduction, an ‘Experimental work’ part, a ‘Results and discussion’ part, and a conclusion. The manuscript is concise but clear although some efforts are needed regarding english spelling.
Overall the results of the study might be of interest to some of the readers of the journal.
Some additions should be made in my opinion though:
- l 66: reference other codes such as ASTM, Eurocodes or FIB bulletins. Please highlight the novelty more clearly explaining the interest of the investigated parameters. As the manuscript is short, I recommend an in-depth literature review
- l 203: could you eventually provide a recommendation irrespective of the fiber length (so that t could maybe be of interest to the standards)
- l 241: please introduce the conclusions
Author Response
Manuscript number: Materials-1378920
MS Type: Article
Title: Study on adaptability of test methods for workability of fresh self-compacting SFRC
Correspondence Author: Xinxin Ding
Dear editor,
Thanks very much for your attention and the reviewers’ evaluation and comments on paper Materials-1378920. Those comments are all valuable and very helpful for revising and improving our paper, as well as the important guiding significance to our researches. Here, we have made extensive modification on the original manuscript, and carefully proof-read the manuscript to minimize typographical, grammatical, and bibliographical errors. The main revised parts are represented as Red. We also attached revised manuscript in the format of MS word for your approval. Here below is our description on revision according to the reviewer 2’ comments.
Comment 1-l 66: reference other codes such as ASTM, Eurocodes or FIB bulletins. Please highlight the novelty more clearly explaining the interest of the investigated parameters. As the manuscript is short, I recommend an in-depth literature review
Response:Thank s for your kind reminder. We have revised the introduction and add one reference [15], and explained the investigated parameters more clearly in line 82-87 of the revised manuscript.
Comment 2-l 203: could you eventually provide a recommendation irrespective of the fiber length (so that t could maybe be of interest to the standards)
Response:It has been added in line 222-225 in the revised manuscript.
Comment 3- l 241: please introduce the conclusions
Response:It has been added in the line 266-270 in the revised manuscript.
Round 2
Reviewer 2 Report
The authors adressed the various issues pointed out by the reviewers and the manuscript could be published in my opinion.